# Analysis of the Variability in Different Criteria to Define the Success of Bariatric Surgery: Retrospective Study 5-Year Follow-Up after Sleeve Gastrectomy and Roux-en-Y Gastric Bypass

**DOI:** 10.3390/jcm12010187

**Published:** 2022-12-26

**Authors:** Sergi Sanchez-Cordero, Amador Garcia Ruiz de Gordejuela, Ramon Vilallonga, Oscar Gonzalez, Ana Ciscar, Andreea Ciudin, Alba Zabalegui, Manuel Armengol

**Affiliations:** 1Metabolic and Bariatric Surgery Unit, Sant Joan Despi Moises Broggi Hospital, Integral Health Consortium, 08970 Sant Joan Despí, Spain; 2Surgery Department, Universitat Autonoma de Barcelona (UAB), 08193 Barcelona, Spain; 3Endocrine-Metabolic and Bariatric Unit, Vall Hebron Barcelona Hospital, 08035 Barcelona, Spain; 4Endocrinology and Nutrition Department, Hospital Universitari Vall d’Hebron, 08193 Barcelona, Spain; 5Diabetes and Metabolism Research Unit, Vall d’Hebron Institut de Recerca (VHIR), Universitat Autonoma de Barcelona, Pg Vall d’Hebron 119–129, 08035 Barcelona, Spain; 6Centro de Investigacion Biomedica en Red de Diabetes y Enfermedades Metabólicas Asociadas (CIBERDEM), Instituto de Salud Carlos III (ISCIII), 28035 Madrid, Spain

**Keywords:** bariatric surgery, success criteria, sleeve gastrectomy, Roux-en-Y gastric bypass

## Abstract

(1) Background: The current criteria for defining good or bad responders to bariatric surgery based on the percentage of weight loss do not properly reflect the therapeutic impact of the main bariatric techniques. At present there is an urgent need to fill this gap and provide scientific evidence that better define the success or failure of bariatric surgery in the long term. (2) Methods: This is a retrospective database study of a prospective cohort with 5-year follow-up. We established the success or failure of bariatric surgery in terms of weight loss according to a selected criterion: (1) Halverson and Koehler; (2) Reinhold modified by Christou; (3) Biron; (4) TWL > 20%; (5) percentage of changeable weight (AWL > 35%). We analyzed sensitivity and specificity for successful weight loss. (3) Results: 223 (38.7%) patients underwent sleeve gastrectomy (LSG) and 353 (61.2%) underwent laparoscopic Roux-en-Y gastric bypass (LRYGBP). The success rates at 5 years are: EWL > 50% 464 (80%), Reinhold 436 (75.6%), Biron 530 (92%), TWL > 20% 493 (85.5%), AWL < 35% 419 (72.7); ≥50% EWL and alterable weight loss AWL > 35 were the most adequate criteria as their specificities and sensibility were far above >80%. (4) Conclusions: The present study shows how the different definitions of success or failure are inconsistent in relation to the outcomes of BS. However, there are some criteria that associate statistically significant differences for the resolution of comorbidities and show the highest sensitivity and specificity rates.

## 1. Introduction

In the last decades, the growing prevalence of obesity and overweight has become an important global social and health problem [1,2]. Obesity is a chronic inflammatory disease that requires a multidisciplinary approach and lifelong therapeutic management [3]. Bariatric surgery (BS) is currently the most effective and efficient therapeutic option for achieving the goals of significant and sustained weight loss, as well as reducing the metabolic comorbidities related to obesity [4,5]. Nevertheless, the rising numbers of bariatric procedures performed worldwide has been followed also by a significant number of patients experiencing insufficient weight loss or weight regain [6,7,8].

To report outcomes, different definitions for success and failure of weight loss after bariatric surgery have been described in the bariatric literature. However, all of them are based on numerical data related to weight and do not consider other variables, such as the evolution of the related diseases or patient’s perspectives. Additionally, all criteria used so far are arbitrary and lack scientific support for the several determined cutoff points.

Furthermore, the current criteria for defining the success and failure of bariatric surgery based on the percentage of weight loss do not reflect the therapeutic success of the main bariatric techniques because it does not always correlate with the resolution of related diseases and/or mortality over time [9,10]. In 2009, Dallal et al. found that excess weight loss (%EWL) was substantially less accurate for expressing bariatric outcome than other methods [11]. Van de Laar et al. published that %EWL is consistently less accurate than %TWL [12,13,14], and these findings have been replicated [15,16]. On the other hand, Corcelles et al. suggested that a total weight loss (%TWL) > 20% in the first year after surgery indicated a “good responder” [17] and was associated with a reduction in cardiovascular risk factors after weight loss (WL) of 5–10% regardless of the type of the medication taken [18].

There is a need to provide scientific evidence that allows us to determine if there is a criterion or combination of criteria that better define the success or failure of BS in the long term.

Given the variability in reported measures and the lack of consensus for the definition of weight loss success after BS, the main goals of this study were to report our experience and to evaluate the variability based on different definitions of success following bariatric surgery of the main primary bariatric surgical techniques (sleeve gastrectomy and gastric bypass) after 5 years of follow-up.

## 2. Materials and Methods

We performed a retrospective database study of a prospective cohort with 5-year follow-up. Patient data were collected retrospectively from patients undergoing bariatric surgery at the metabolic and bariatric surgery unit of the Vall d’Hebron University Hospital in Barcelona, Spain.

As per protocol at our site, the inclusion criteria were: (a) BMI > 40 kg/m^2^ or BMI > 35 kg/m^2^ with at least one obesity-related comorbidity, (b) age between 18 and 65 years, (c) patients that underwent Roux-en-Y-gastric bypass or sleeve gastrectomy with a completed 5-year follow up at our site. Exclusion criteria were: (a) open or robotic surgery, (b) other bariatric techniques, and previous bariatric surgery, for any reason (e.g., conversion of sleeve gastrectomy to Roux-en-Y-gastric bypass for complications of SG).

All procedures were performed laparoscopically following a technique standardized by the team of surgeons [19].

All patients were evaluated by a multidisciplinary team, which included surgeons, endocrinologists, dietitians, endoscopists, radiologists, anesthesiologists, psychologists, and specialized nursing staff, to assess the criteria for bariatric surgery. Preoperative evaluation included abdominal ultrasound, gastroscopy, laboratory analysis and nutritional status.

Patient demographic data (age, weight, BMI), related diseases (hypertension (HTN), diabetes mellitus 2 (DM2), dyslipemia (DLP), arthritis, heart disease, vasculopathy and hiatal hernia, obstructive apnea syndrome (OSA)), weight, resolution of related diseases, and mortality during the 5 years of follow-up were collected. The DM2 definition was based on American Diabetes Association (ADA) criteria: HbA1c > 6.5%, FBG > 126 mg/dL, glycemia 2 h after a standardized oral glucose tolerance test >200 mg/dL, or random blood glucose test >200 mg/dL. Hypertension (HTN) was defined as blood pressure >120/80 mmHg and/or antihypertensive treatment. OSAS was defined as >5 apnea–hypoapnea respiratory disturbance per hour and/or CPAP treatment. Dyslipidemia (DLP) was defined as total cholesterol >200 mg/dL and/or triglycerides >150 mg/dL and/or hypolipemic treatment. We consider related disease remission when the patient did not need medication with adequate management. We defined improvement as a reduction in drug dose for the treatment of weight-related disorders.

Weight progression was assessed in terms of BMI, weight loss, expressed (1) as a percentage of excess body mass index loss (% EBMIL), (2) percentage of total weight loss (% TWL), and (3) percentage of changeable weight loss (AWL%), described by Van der Laar [20]. Finally, the success rates and weight regain, and the impact of associated comorbidities, morbidity, and mortality of both techniques were analyzed.

(1)% EBMIL: (initial BMI − current BMI)/(initial BMI − 25) × 100(2)% TWL: (initial weight − current weight)/initial weight × 100(3)% AWL: final BMI/(initial BMI − 13) × 100

We established the success or failure of bariatric surgery in terms of weight loss according to a selection of criteria. The five criteria selected were: (a) Halverson and Koehler, which defines surgery success as >50% excess weight loss [21]; (b) Reinhold modified by Christou that defines failure as a final BMI greater than 35 kg/m^2^ [22]; (c) Biron, who defines success as a final BMI less than 35 if preoperatively the initial BMI is less than 50 kg/m^2^, or a final BMI less than 40 kg/m^2^ if preoperatively the initial BMI is greater than 50 kg/m^2^ [23]; (d) Grover et al. proposed 20% TWL as success of the surgery [24]; and (d) percentage of changeable weight (AWL > 35%), as described by Van der Laar [25] (Table 1).

To validate the classic criteria for bariatric weight loss success, a benchmark is needed to compare, BMI baseline independently, the outcomes of BS. From these five criteria, AWL and TWL are the two metrics least influenced by differences in baseline BMI. Our results showed that AWL > 35% was the most restrictive criterion with the highest specificity and based on our criteria the most accurate to use as a gold standard benchmark.

We expressed sensitivity and specificity for the five criteria for successful weight loss as percentages and calculated them with the number of false positive results (successful according to the criterion) and false negative results (unsuccessful according to the criterion). We consider real success the patients after 5 years with AWL > 35 criterion positive or TWL > 20%. We considered a criterion inadequate if either sensitivity or specificity was below 60%. We considered a criterion useful if both sensitivity and specificity were above 75%.

Statistical analysis was performed with SPSS (v23.0, IBM Corporation, Armonk, NY, USA). Quantitative variables that followed a normal distribution were summarized by means and standard deviations. For non-Gaussian variables, the median and range were used. Qualitative variables were summarized by number and percentage of cases; *p*-values < 0.05 were considered significant.

## 3. Results

A total of 919 bariatric surgeries were performed at our site between January 2006 and December 2014, of which 861 were Roux-en-Y-gastric bypass and sleeve gastrectomy.

There were 576 cases (66.9%) that met the inclusion criteria and were eligible for our study: 223 (38.7%) patients underwent sleeve gastrectomy (LSG) and 353 (61.2%) underwent laparoscopic Roux en Y gastric bypass (LRYGBP).

The baseline characteristics (demographics, anthropometry, obesity-related diseases) are summarized in Table 1. There were no significant differences in age, sex, preoperative weight, and BMI, nor distribution of related diseases (DM2, HTN, OSAS, DLP, arthropathy, heart disease, vascular disease, and fibromyalgia) between groups, except for hiatal hernia (*p* < 0.001) and steatosis (*p* > 0.001), which were significantly more predominant in the LRYGB group (Table 2).

### 3.1. Postoperative Weight Monitoring

The postoperative weight loss was collected prospectively during the follow-up medical visits. Overall, the mean BMI preoperative was 45.44 ± 5.67 kg/m^2^, reaching the nadir BMI at 18 months with a mean BMI of 29.3 ± 4.6. 5 kg/m^2^. 5 years follow-up the mean BMI was 31.5 ± 5.6 and %TWL 30.9 ± 11.

Mean preoperative BMI was 46.10 ± 6.9 kg/m^2^ for the LSG group. The %TWL 5 years after surgery was 31.8. Five years after surgery, the mean BMI was 33.2 ± 6.2 kg/m^2^ with %EBMIL of 62.52% and %TWL of 28.1 for the LSG group. Preoperative mean BMI was 45.1 ± 4.4 kg/m^2^ for the LRYGB group. At 5-year follow-up, the BMI after LRYGB was 30.6 ± 4.9 kg/m^2^, with %EBMIL of 74% for LRYGB and 33.3 ± 10.5 %TWL. There are statistically significant differences between the LSG and LRYGB groups at five years after surgery (Figure 1 and Figure 2).

### 3.2. Success Rates

Figure 3 shows the anthropometric data collected and BMI evolution during the 5-year follow-up. According to the several success criteria proposed in the study, a total of 369 (64%) patients met some success criteria with a mean BMI of 28.5 kg/m^2^ vs. 37.03 kg/m^2^ at 5-year follow-up. The success rates at 5 years were: EWL > 50% 464 (80%), Reinhold 436 (75.6%), Biron 530 (92%), TWL > 20% 493 (85.5%), and AWL < 35% 419 (72.7%) (Figure 3). For the success group, there were statistically significant differences in the evolution of related diseases: HT 61.6 vs. 47.8 (*p* < 0.05); DM2 40.2 vs. 28.2 (*p* < 0.05), and OSAS 77.7% vs. 65.4 (*p* < 0.05).

Table 3 show the success criteria rates for the different techniques performed. Table 3 is for patients in the LSG group. The highest rates for LSG groups are for the Biron (85%) and TWL (78%) criteria. The preoperative BMI is 44.4 and 46.2, and at 5-year follow-up, 31.3 and 31.6, respectively. Table 4 is for patients in the RYGB group, where higher success rates are calculated. In this group, higher success rates are obtained by the Biron (96%), TWL (90%), and EWL (88%) criteria.

### 3.3. Sensitivity and Specificity

Sensitivity and specificity for the five criteria for weight loss success is based on the 576 patients beyond five years and with reference to success with AWL and TWL criteria, as shown in Figure 4. Applying AWL > 35% as the definition of the real success, Biron, Grover, and Reinhold criterion modified by Christou could be considered inadequate, as their specificities were (far) below 70%. They all had low specificities, leaving too many poor responders unnoticed. Weight loss criteria ≥ 50%EWL were adequate in the sleeve and BMI > 50 kg/m^2^ groups, as their sensitivities and specificities were all above 70%.

In Figure 5, we expressed the results of the sensitivity and specificity analysis for TWL > 20% as the definition of the real success. Weight loss criteria ≥ 50%EWL and alterable weight loss >35% were the most adequate criteria as their specificities and sensibility were far above >80%. Alterable weight loss presented the highest specificity rates in all the subgroup analysis. Biron criteria showed good specificity and sensibility in the stratified analysis per surgery technique but worse results in the stratified analysis per initial BMI.

## 4. Discussion

Obesity is a chronic relapsing disease and at present there is no clear average weight loss outcome to define success in bariatric surgery. In the present study, the weight loss and success rates of a retrospective cohort in a tertiary care setting series are analyzed. The weight loss evolution trends are similar to those previously described in the literature [10] between SG and LRYGB. Nevertheless, high variability rates are observed for the different selected criteria. For the same study population, we demonstrated that using different definitions for good responders showed success rates ranging from 62% to 96%.

The lack of consensus and the given variability of success rates 5 years after surgery do not allow a reliable comparison between different series reported in the literature. For this reason, a consensus that unifies the success criteria is needed.

Although definitions based on changes in BMI have a good correlation between clinical outcomes and the risk of developing comorbidities, it is unclear which of the cutoff changes in BMI is required to achieve clinical goals [20]. Additionally, the increase in BMI or being above a certain BMI has not been clearly correlated with the recurrence of comorbidities. In 2007, Deitel et al. [26] recommended publishing results in bariatric surgery as excess BMI lost (%EBMIL). However, this criterion is based on an ideal weight for a BMI of 25 Kg/m^2^, which has significant limitations and does not reflect properly the normal weight of the patients.

On the other hand, the %EWL is the most used parameter in bariatric surgery outcomes. The difficulty lies in determining the ideal weight for a specific individual [12]. This is the reason many authors recommend the TWL or AWL as an independent BMI indicator of the ideal weight [17].

The inconsistency between the different definitions in the literature to define responders was already described by Bonouvrie et al. [9]. For this reason, we used definitions influenced by the initial weight, such as the EWL or the final BMI, and others that are completely independent, such as the AWL or the TWL. Nevertheless, at present, the cutoff of %TWL needed to be considered successful surgical weight loss is still controversial. In consequence, what cutoff of percentage weight regain should be considered significant is also controversial.

In our series, there is variability between the different criteria for success and failure of bariatric surgery. Globally, the rates range between 73% and 92% for the different classification criteria. All the BMI-dependent criteria initially present higher success rates than AWL. We believe that the classic criteria may be overestimating the favorable results of the BS success rate. Therefore, EWL, BMI, and Biron criteria, despite having a high success rate, do not show the correlation with the related disorder studied. However, AWL shows a good correlation with statistically significant differences between the success rates and related diseases as HTN (*p* < 0.0001), DM2 (*p* < 0.0001), and DLP (*p* < 0.005).

At present, there is no gold standard bariatric surgery (except for LRYGB) to compare groups and population. In addition, different techniques (SG, LRYGB, SADI-S, or even duodenal switch (DS)) have different goals and weight loss patterns, and the long-term effect after BS could be dilucidated by the lifestyle habits and the presence of complications [27].

On the other hand, we showed a marked difference in the success rates between the different techniques studied. Gastric bypass presents better success rates than sleeve gastrectomy in all the criteria studied.

For the first time, an attempt to validate the bariatric weight loss criteria for success was made. All the classic bariatric criteria and the two most popular were compared in a large single cohort of patients after sleeve gastrectomy and Roux-en-Y gastric bypass. Results were disappointing for Biron and Reinhold criteria for their low specificity and sensibilities attending both analyses.

As the criteria tests are widely used by bariatric professionals and researchers, they represent a consensus. The specificity rates for the Biron and Reinhold criteria were very slow for different techniques and BMI at baseline, so the criteria should be abandoned all together. Both are criteria that depend on BMI at baseline, so it is important to inform the patients preoperative on the real expectations. The results of our study suggest that classic static criteria based on BMI at baseline are not appropriate for long-term results. In our cohort, only the EWL > 50% and AWL > 35 criteria are adequate to classify bariatric outcome with >90% sensitivity and specificity attending different techniques and BMI at baseline.

This study again proves, similar to previous studies [14,15], the variability of the successful results for each criteria. As explained in previous studies, although bariatric patients typically show a wide variety of BMI at baseline, their weight loss essentially is baseline BMI-independent. We strongly believe that the success criteria used need to be BMI-independent as well.

Finally, bariatric surgery aims to improve cardiovascular risk associated with metabolic syndrome and life expectancy in our patients, as well as to improve their quality of life [23]. None of the definitions with greater global acceptance include any of these factors. However, in our study, we have shown that the Halverson and Koehler criteria and TWL >20% significantly evaluated the resolution of obesity-related comorbidities as well. The success of bariatric surgery should be measured by postoperative weight loss, improvement of metabolic related diseases, impact on quality of life, and side effects caused by the type of surgery.

Success of a bariatric procedure should be measured, in both the short term and long term, by postoperative weight loss, improvement of metabolic comorbidities, and impact on quality of life. Since it is difficult to achieve long-term results on the quality of life, the challenge is to correlate bariatric and metabolic surgery success in the long term at the nadir BMI.

## 5. Conclusions

The present study shows how the different definitions of success or failure are inconsistent in relation to the outcomes of BS. However, there are some criteria that associate statistically significant differences for the resolution of comorbidities and show the highest sensitivity and specificity rates.

## Figures and Tables

**Figure 1 jcm-12-00187-f001:**
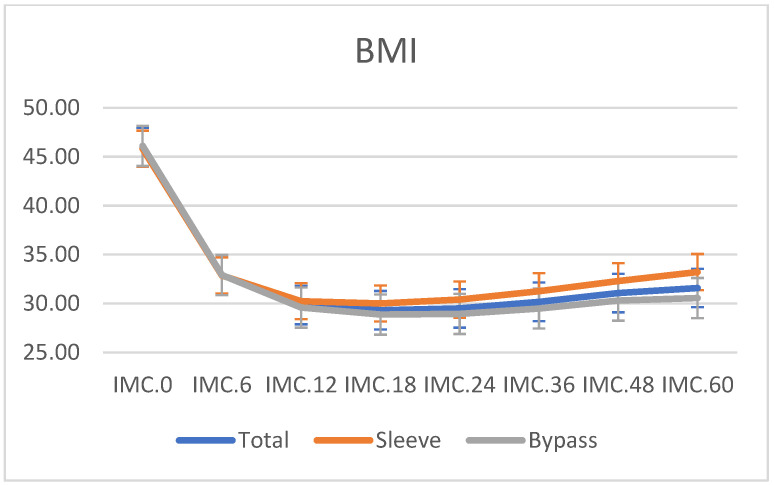
Evolution of body mass index (BMI) during the follow-up of RYGB and SG expressed in months.

**Figure 2 jcm-12-00187-f002:**
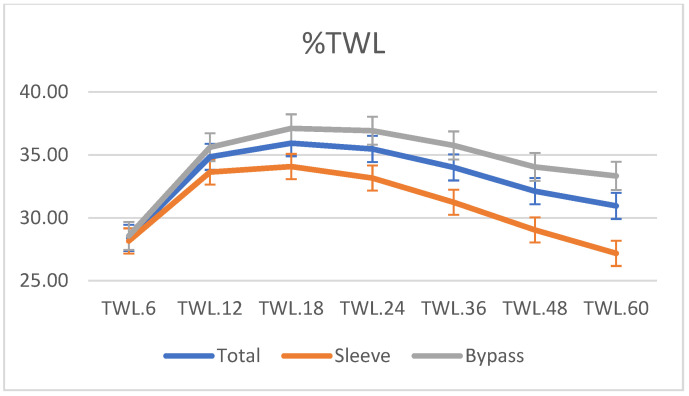
Evolution of % total weight loss (TWL) during the follow-up of RYGB and SG expressed in months.

**Figure 3 jcm-12-00187-f003:**
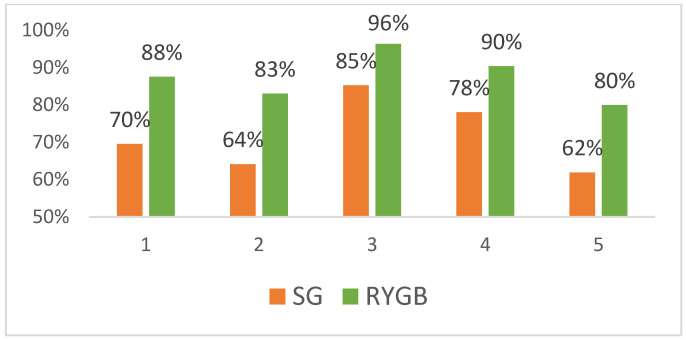
Success rates attending different criteria at 5-year follow-up for LSG and RYGB. (1) EWL > 50%; (2) BMI < 35; (3) Biron criteria is defined by BMI < 35 if the initial BMI < 50 kg/m^2^ and BMI < 40 kg/m^2^ if the initial BMI > 40; (4) TWL > 20%, and (5) AWL > 35%.

**Figure 4 jcm-12-00187-f004:**
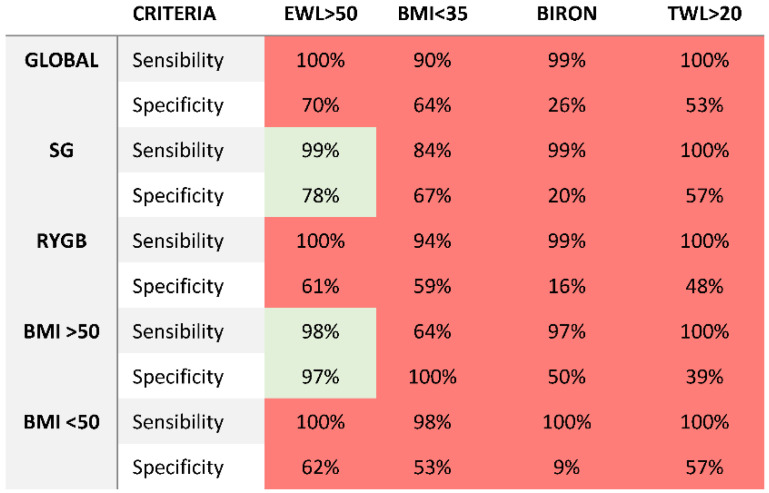
Sensitivity and specificity (expressed as percentages) of 4 criteria for weight loss success. Green: useful criteria with a sensitivity and specificity >70%. Red: inadequate criteria with sensitivity and specificity or specificity <70%.

**Figure 5 jcm-12-00187-f005:**
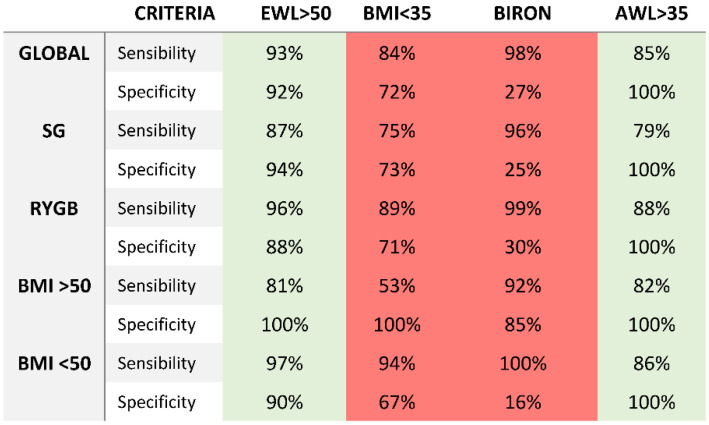
Sensitivity and specificity (expressed as percentages) of 4 criteria for weight loss success. Green: useful criteria with a sensitivity and specificity >70%. Red: inadequate criteria with sensitivity and specificity or specificity <60%.

**Table 1 jcm-12-00187-t001:** Success and failure criteria according to literature reports.

	Halverson and Koehler %EWL	Reinhold Modifed by Christou BMI	Biron et al.	Van der Laar AWL >35%	Grover et al. TWL <20%
Success	>50%	<35	<35 if initial BMI < 50 <40 if initial BMI > 50	>35%	>20%
Failure	<50%	>35		<35%	<20%

**Table 2 jcm-12-00187-t002:** Clinical characteristics of the study participants at baseline. Data are expressed as mean ± SD or N number of individuals and % proportion. BMI.O, BMI before surgery; SG, sleeve gastrectomy; RYGB, Roux-en-Y gastric bypass; HTN, hypertension; DM2, diabetes mellitus type 2; OSAS, obstructive apnea syndrome; DLP, dyslipidemia.

	Total (N = 576)		SG (N = 223)		RYGB (N = 353)		*p*-Value
Age (mean, SD)	45.7	10.3	45.09	11.127	46.07	9.771	0.286
Sex (female, %)	452 (78.5%)		165 (73.9%)		287 (81.3%)		0.058
BMI.0 (kg/m^2^; mean, SD)	45.99	5.82	45.81	6.82	46.10	5.10	0.584
DM2 (n, %)	176 (30.6)		62 (35.2)		114 (64.8)		0.295
HTN (n, %)	291 (50.5)		115 (39.5)		176 (60.5)		0.753
DLP (n, %)	234 (40.6)		98 (41.9)		136 (58.1)		0.229
OSAS (n, %)	390 (67.7)		151 (38.7)		239 (61.3)		1.000
Artropathy (n, %)	272 (47.2)		97 (35.7)		175 (64.3)		0.181
Vasculopaty (n, %)	201 (34.9)		73 (36.3)		128 (63.7)		0.438
Hiatal hernia (n, %)	132 (22.9)		34 (25.8)		98 (74.2)		0.001
Steatosis (n, %)	30 (5.2)		5 (16.7)		25 (83.3)		0.001
Heart disease (n, %)	30 (5.2)		16 (53.3)		14 (46.7)		0.135

**Table 3 jcm-12-00187-t003:** Success rates for different criteria at 5-year follow-up for sleeve gastrectomy interventions. Data are reported as mean and 95% confidence interval (CI 95%) or *n* (%). N, number of individuals; BMI.O, BMI before surgery; BMI.nadir, BMI at nadir post-bariatric surgery; BMI.60, BMI at 5-year post-bariatric surgery; HTN, hypertension; DM2, diabetes mellitus type 2; OSAS, obstructive apnea syndrome; DLP, dyslipidemia.

	EWL > 50% (N = 155)			BMI < 35 (N = 143)				Biron (N = 190)				TWL > 20 (N = 174)				AWL > 35 (N = 137)			
	N/Mean	(%/IC95%)		N/Mean		(%/IC95%)		N/Mean		(%/IC95%)		N/Mean		(%/IC95%)		N/Mean		(%/IC95%)	
Age	44.24	42.4	46.04	44.9	43.08		46.8	45.1	43.5		46.76	44.5	42.8		46.2	43.05	41.1		44.9
Sex(females)	113	72.9%		108		75.5%		143		75.2%		131		75.2%		103		75.1%	
BMI.0	45.0	43.9	46.1	43.05	42.15		43.9	44.4	43.6		45.33	46.2	45.1		47.2	45.66	44.4		46.8
HTN	77	47.7%		71		49.6%		98		50.2%		85		48.9%		60		46.4%	
DM2	35	22.5%		35		24.4%		50		29.8%		39		28.2%		24		25.7%	
OSAS	100	64.5%		92		64.3%		122		66.4%		115		66.3%		89		65.5%	
DLP	63	40.6%		62		43.3%		86		40.6%		71		39.6%		52		37.9%	
BMI.Nadir	27.23	26.63	27.8	26.5	26.03		27.05	27.9	27.37		28.49	28.15	27.4		28.9	27.17	26.4		27.8
BMI.60	30.3	29.65	31.0	29.4	28.8		29.95	31.3	30.7		31.99	31.63	30.8		32.4	30.15	29.3		30.9

**Table 4 jcm-12-00187-t004:** Success rates for different criteria at 5-year follow-up for Roux-Y-gastric bypass interventions. Data are reported as mean and 95% confidence interval (CI 95%) or *n* (%). N, number of individuals; sex is expressed by the number of female patients. BMI.O, BMI before surgery; BMI.nadir, BMI at nadir post-bariatric surgery; BMI.60, BMI at 5-year post-bariatric surgery; HTN, hypertension; DM2, diabetes mellitus type 2; OSAS, obstructive apnea syndrome; DLP, dyslipidemia.

	EWL > 50% (N = 309)			BMI < 35 (N = 293)				Biron (N = 340)				TWL > 20 (N = 319)				AWL > 35 (N = 282)			
	N/Mean	(%/IC95%)		N/Mean		(%/IC95%)		N/Mean		(%/IC95%)		N/Mean		(%/IC95%)		N/Mean		(%/IC95%)	
Age	45.8	44.7	46.9	45.5	44.3		46.6	45.9	44.9		46.9	46.02	44.9		47.09	45.4	44.2		46.6
Sex(females)	259	83.8%		243		82.9%		279		82%		268		84%		234		82.9%	
BMI.0	46.1	45.5	46.7	45.4	44.9		45.97	45.9	45.4		46.5	46.3	45.8		46.9	46.4	45.8		47.04
HTN	148	47.8%		140		48.4%		168		49.4%		156		48.9%		133		46.4%	
DM2	96	28.2%		94		29.6%		108		31.7%		100		31.3%		86		25.7%	
OSAS	203	65.3%		193		65.4%		230		67.6%		212		66.4%		187		65.5%	
DLP	117	38.8%		112		39.9%		129		37.9%		124		38.8%		108		37.9%	
BMI.Nadir	26.9	26.5	27.4	26.5	26.1		26.8	27.3	26.8		27.7	27.1	26.7		27.6	26.7	26.2		27.2
BMI.60	29.4	28.9	39.8	28.9	28.5		29.3	30.1	29.6		30.5	29.7	29.3		30.25	29.06	28.6		29.5

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
