# Peer review of "Analysis of the Variability in Different Criteria to Define the Success of Bariatric Surgery: Retrospective Study 5-Year Follow-Up after Sleeve Gastrectomy and Roux-en-Y Gastric Bypass"

_jcm, 2022, doi:10.3390/jcm12010187_

Round 1

Reviewer 1 Report

Sanchez-Cordero et al analyzed a large cohort of bariatric surgery (BS) patients who underwent either sleeve gastrectomy (SG) or Roux en Y gastric bypass (RYGB) to investigate which are the appropriate criteria that should be used to define success. This is a very interesting approach that can provide valuable information about the optimal way to evaluate the outcome of BS procedures.

Unfortunately, the manuscript lacks clarity and needs significant editing before publication. Moreover, the presentation of the results is problematic and the conclusions are not fully supported by the results.

Specific points:

1. Figures 1 and 2 should depict the standard deviation (error bars). Is there a significant difference between the percentage of total weight loss between SG and RYGB groups?

2. Tables 3a and 3b should also include the non-successful operations data and a side-by-side statistical comparison between successful and non-successful groups to analyze differences in the evolution of related diseases. Does sex mentioned in the table refer to males or females?

3. Figures 4 and 5 should be presented in English and be consistent (green and red only colors). What is the rationale behind the selection of AWL>35 as benchmark?

4. Steatosis is also significantly different between RYGB and SG groups (Table 2)

5. References should be included properly in the discussion section and all the results should be presented in the appropriate section and not in the methods. Moreover, the whole manuscript needs significant language and format editing.

Author Response

Specific points:

1.Figures 1 and 2 should depict the standard deviation (error bars).

Modification on figure 1 and 2 is made

Is there a significant difference between the percentage of total weight loss between SG and RYGB groups?

There is significant sadistically difference. There is a modification made in relation to this point. Line 163, 164 paragraph 3 page 5

  1. Tables 3a and 3b should also include the non-successful operations data and a side-by-side statistical comparison between successful and non-successful groups to analyze differences in the evolution of related diseases.

This is a very interesting point, the subanalysis is made by group but there was not satistical significant differences and the general idea of the paper was elucidated. It could be a great analysis on a following paper.

Does sex mentioned in the table refer to males or females?  

Females

  1. Figures 4 and 5 should be presented in English and be consistent (green and red only colors).

The figures 4 and 5 are fully revised with the appropriate considerations.

What is the rationale behind the selection of AWL>35 as benchmark?

First, AWL and TWL are the benchmark because to date there are the only two successful criteria BMI independent. In addition, there are some authors recommending the use of TWL or AWL as an indicator of success (reference 17).

After analyze both metrics, we described the following sentence on page 4, first paragraph line 119-123: Our results showed that AWL>35 was the most restrictive criteria with the highest specificity and based on our criteria the most accurate to use as a gold standard benchmark.

  1. Steatosis is also significantly different between RYGB and SG groups (Table 2).

Although, there is a sadistically difference in the preoperative data, we think there is not a clinical consideration or an implication on the outcome for the final analysis on the selected criteria to define success..

Modification on the text is made in relation to this consideration.  Line 146-147 paragraph 5 page 4

  1. References should be included properly in the discussion section and all the results should be presented in the appropriate section and not in the methods.

The revision is made, and all the references are revised in the discussion section.  Comments on results of the studies include in the references are avoid.

Reviewer 2 Report

Congratulation for well written paper, defining success and failure has been difficult, this report gives light on these definitions. I believe the definition of success should be divided in two groups according to the starting BMI. Patient with BMI>55 and < 55, weight loss in these two groups is different and resolution of comorbidity variable, patients' self-satisfaction with weight loss is important  

Author Response

Thank you very much for this kind review. It is a very interesting point to analyze the differences between BMI >55 and <55 and we are working on this future line to show better results on the future.

Round 2

Reviewer 1 Report

The revised version of the manuscript addressed most of my concerns and is improved. However, the manuscript still needs significant editing before publication because the present form is quite difficult to read.

Author Response

Dear editor, 

I appreciate you taking the time to review our manuscript. The full text is revised by an English native speaker in order to improve the language style and the comprehension. I would like this final version will be able to be published in the journal.  
